# Insights into electrosensory organ development, physiology and evolution from a lateral line-enriched transcriptome

Melinda S Modrell[1], Mike Lyne[2,3], Adrian R Carr[2,3], Harold H Zakon[4,5], David Buckley[6,7], Alexander S Campbell[1], Marcus C Davis[8], Gos Micklem[2,3], Clare VH Baker[1]*

[1]Department of Physiology, Development and Neuroscience, University of Cambridge, Cambridge, United Kingdom; [2]Cambridge Systems Biology Centre, University of Cambridge, Cambridge, United Kingdom; [3]Department of Genetics, University of Cambridge, Cambridge, United Kingdom; [4]Department of Neuroscience, The University of Texas at Austin, Austin, United States; [5]Department of Integrative Biology, The University of Texas at Austin, Austin, United States; [6]Departmento de Biodiversidad y Biología Evolutiva, Museo Nacional de Ciencias Naturales-MNCN-CSIC, Madrid, Spain; [7]Department of Natural Sciences, Saint Louis University - Madrid Campus, Madrid, Spain; [8]Department of Molecular and Cellular Biology, Kennesaw State University, Kennesaw, United States

*For correspondence: cvhb1@cam.ac.uk

Competing interests: The authors declare that no competing interests exist.

**Abstract** The anamniote lateral line system, comprising mechanosensory neuromasts and electrosensory ampullary organs, is a useful model for investigating the developmental and evolutionary diversification of different organs and cell types. Zebrafish neuromast development is increasingly well understood, but neither zebrafish nor *Xenopus* is electroreceptive and our molecular understanding of ampullary organ development is rudimentary. We have used RNA-seq to generate a lateral line-enriched gene-set from late-larval paddlefish (*Polyodon spathula*). Validation of a subset reveals expression in developing ampullary organs of transcription factor genes critical for hair cell development, and genes essential for glutamate release at hair cell ribbon synapses, suggesting close developmental, physiological and evolutionary links between non-teleost electroreceptors and hair cells. We identify an ampullary organ-specific proneural transcription factor, and candidates for the voltage-sensing L-type $Ca_v$ channel and rectifying $K_v$ channel predicted from skate (cartilaginous fish) ampullary organ electrophysiology. Overall, our results illuminate ampullary organ development, physiology and evolution.

## Introduction

The lateral line system of fishes and aquatic amphibians is a good model for studying the diversification of different organs and cell types, both in development and evolution. In jawed vertebrates, this sensory system ancestrally includes mechanosensory neuromasts and electrosensory ampullary organs ('ampullae of Lorenzini'), both of which develop - together with their afferent neurons - from individual embryonic lateral line placodes (*Northcutt et al., 1995*; *Modrell et al., 2011a*; *Gillis et al., 2012*). (The jawless lampreys have neuromasts and electroreceptors, but the latter are collected in 'end buds' at the surface, rather than recessed in ampullary organs, and their embryonic origin is unknown; the lateral line system of hagfishes, which lack electroreceptors altogether, is

secondarily reduced; *Braun, 1996*; *Braun and Northcutt, 1997*.) The electrosensory division of the lateral line system was lost independently in the lineages leading to extant neopterygian fishes (gars, bowfin and teleosts) and to anuran amphibians (neither of the major anamniote lab models, i.e., the teleost zebrafish and the frog *Xenopus*, has electroreceptors). However, electrosensory lateral line organs evolved independently at least twice within the teleosts, most likely from neuromast hair cells (*Bullock et al., 1983*; *Northcutt, 1986*; *Bodznick, 1989*; *Alves-Gomes, 2001*; *Bodznick and Montgomery, 2005*; *Kawasaki, 2009*; *Baker et al., 2013*). The entire lateral line system was lost in amniotes, with the transition to life on land.

The loss of the electrosensory division of the lateral line system in different vertebrate lineages shows that ampullary organ development must be genetically separable from neuromast development. Indeed, even within the same lateral line placode-derived sensory ridge, neuromasts form first, along the center of the ridge, while ampullary organs form later, on the flanks (*Schlosser, 2002*; *Northcutt, 2005a*; *Piotrowski and Baker, 2014*). Neuromasts and ampullary organs are morphologically distinct: neuromasts contain mechanosensory hair cells, plus supporting cells that secrete a gelatinous cupula; ampullary organs comprise a sensory epithelium of electroreceptor and supporting cells located at the base of a duct filled with conductive jelly, leading to a surface pore (*Northcutt, 1986*; *Jørgensen, 2005*; *Baker et al., 2013*). Ampullary electroreceptor cells generally have an apical primary cilium but either no or few apical microvilli, which are not organized into the 'hair bundle' (stair-case array) that characterizes hair cells (*Northcutt, 1986*; *Jørgensen, 2005*; *Baker et al., 2013*). The molecular mechanisms underlying neuromast formation from the migrating posterior lateral line primordium in zebrafish have been intensively studied (*Chitnis et al., 2012*; *Piotrowski and Baker, 2014*; *Thomas et al., 2015*), but our molecular understanding of ampullary organ development is very limited.

Like hair cells (and also retinal and pineal photoreceptors, and retinal bipolar neurons), all vertebrate electroreceptors have electron-dense pre-synaptic bodies surrounded by a large pool of synaptic vesicles (*Northcutt, 1986*; *Jørgensen, 2005*). Such 'ribbon synapses' respond to graded signals and are capable of sustained neurotransmitter release (*Matthews and Fuchs, 2010*; *Pangršič et al., 2012*; *Safieddine et al., 2012*; *Nicolson, 2015*; *Wichmann and Moser, 2015*; *Moser and Starr, 2016*). While the specific proteins involved in ribbon synapse function are increasingly understood in retinal photoreceptors and hair cells (*Matthews and Fuchs, 2010*; *Pangršič et al., 2012*; *Safieddine et al., 2012*; *Nicolson, 2015*; *Wichmann and Moser, 2015*; *Moser and Starr, 2016*), all that is known about neurotransmission at non-teleost electroreceptor ribbon synapses is from work in dissected skate (cartilaginous fish) ampullary organs, which showed that activation of L-type voltage-gated calcium channels results in the release of a 'glutamate-like' neurotransmitter (*Bennett and Obara, 1986*).

Similarly, our only detailed understanding of non-teleost ampullary organ physiology until very recently had come from current- and voltage-clamp approaches to study epithelial currents in dissected single ampullary organ preparations from skates (*Bennett and Obara, 1986*; *Lu and Fishman, 1995*; *Bodznick and Montgomery, 2005*). These revealed that L-type voltage-gated calcium channels are required both for voltage-sensing in the apical (lumenal, i.e., exterior-facing) electroreceptor membrane, and for neurotransmitter release basally. The basal membrane is repolarized by voltage-gated potassium channels and calcium-dependent chloride channels, while the apical membrane is repolarized by the calcium-gated potassium channel BK (*Bennett and Obara, 1986*; *Lu and Fishman, 1995*; *Bodznick and Montgomery, 2005*; *King et al., 2016*). BK was recently cloned directly from skate ampullary organs (*King et al., 2016*), a little over 40 years after its properties were initially discovered using the same preparation (*Clusin et al., 1975*; *Clusin and Bennett, 1977a*; *Clusin and Bennett, 1977b*). While the current manuscript was under review, whole-cell patch-clamp experiments on dissociated electroreceptors from adult skates, together with transcriptome profiling, revealed that the L-type voltage-gated calcium channel Ca$_v$1.3 mediates the low-threshold voltage-activated inward current, and works together with BK to mediate electroreceptor membrane oscillations (*Bellono et al., 2017*). Other than this recent exciting advance (*Bellono et al., 2017*), the specific ion channels and subunits involved in electroreceptor function have not been identified in any vertebrate.

In short, our understanding of the specific molecular basis of both vertebrate electroreceptor development and physiology is still rudimentary. The few candidate gene approaches reported thus far have identified some transcription factors, and a few other genes, expressed in both ampullary

organs and neuromasts in larval axolotl (*Metscher et al., 1997*; *Modrell and Baker, 2012*), paddlefish (*Modrell et al., 2011a*, *2011b*; *Butts et al., 2014*), shark and skate (*Freitas et al., 2006*; *Gillis et al., 2012*). However, a candidate gene approach will not identify genes important specifically for ampullary organ development or function. Here, we report an unbiased transcriptomic approach in paddlefish (a non-teleost chondrostean fish) to identify such genes, which has yielded novel and wide-ranging insights into the development, physiology and evolution of non-teleost ampullary organs.

## Results

We generated transcriptomes at stage 46 (the onset of independent feeding; *Bemis and Grande, 1992*) from pooled paddlefish opercula (gill-flaps), which are covered in ampullary organs plus some neuromasts, versus fins, which have a generally similar tissue composition but no lateral line organs at all (*Modrell et al., 2011a*, *2011b*). Differential expression analysis yielded 490 genes, excluding duplicates, enriched at least 1.85-fold (log$_2$fold 0.89) in operculum versus fin tissue, hereafter designated 'lateral line-enriched' (*Supplementary file 1*). Of these genes, 112 were uncharacterized loci or could only be assigned to a protein family, while a further 44 were described as being 'like' a specific gene, leaving 334 assigned genes (*Supplementary file 1*). *Figure 1* shows a molecular function analysis using Gene Ontology (GO) terms, where available.

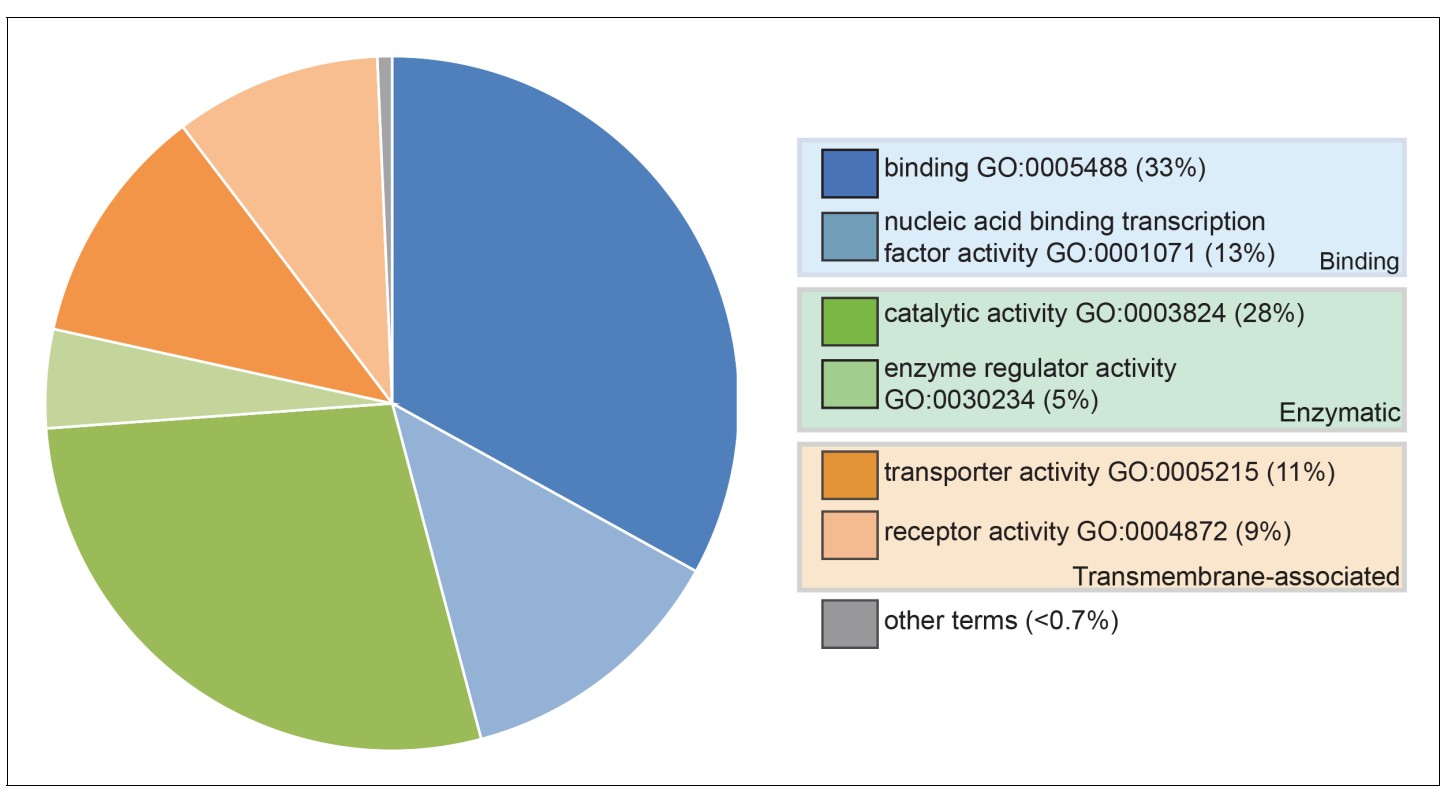

**Figure 1.** Pie chart showing the results of a PANTHER classification analysis by molecular function of 332 transcripts from the paddlefish lateral line-enriched dataset with associated gene ontology (GO) terms. The percentage of function hits is indicated in parentheses. Binding activities GO:0005488 (blue) represent ~39% of the function hits (16% nucleic acid binding GO:0003637; 17% protein binding GO:0005515; 6% other binding, including calcium and lipid binding). Catalytic activities GO:0003824 (green) account for ~35% of the function hits (4% enzyme regulator activity GO:0030234; 9% transferase; and 11% each hydrolase and other catalytic activities). Transmembrane-associated activities (orange) represent ~19% of function hits (13% transporter GO:0005215; 6% receptor GO:0004872). The remaining ~7% of function hits are comprised of signal transducer GO:0004871, structural molecule GO:0005198 and antioxidant GO:0016209 activities.

## Conservation in ampullary organs of transcription factors critical for mechanosensory hair cell development

The lateral line-enriched dataset (*Supplementary file 1*) includes several transcription factor genes whose expression we had previously reported in both ampullary organs and neuromasts in larval paddlefish, suggesting that the differential expression analysis had been successful. These were: the homeodomain transcription factor genes *Six1* and *Six2* (*Modrell et al., 2011a*); the HMG-domain *SoxB1* class transcription factor gene *Sox3* (*Modrell et al., 2011b*); and the basic helix-loop-helix (bHLH) transcription factor gene *Atoh1*, whose lateral line organ expression at stages 40–45 was noted, in passing, in a study on cerebellum development (*Butts et al., 2014*).

Atoh1 is essential for hair cell formation (*Bermingham et al., 1999*). Paddlefish *Atoh1* is 18.6-fold lateral line-enriched (*Supplementary file 1*), and expressed by stage 33 within the otic and preopercular neuromast canal lines (*Figure 2A*). It is expressed within most canal lines and the migrating posterior lateral line primordium by stage 37 (*Figure 2B*), and in the developing ampullary organ fields by stage 39 (*Figure 2C*), where it persists at stage 46 (*Figure 2D*).

In mouse cochlear explants, Six1 and its co-factor Eya1, acting cooperatively with SoxB1 subfamily member Sox2, are sufficient to induce *Atoh1* (*Ahmed et al., 2012*). Like *Six1*, *Eya1* is also expressed in paddlefish ampullary organs (*Modrell et al., 2011a*), while *Sox2* is 3.3-fold lateral line-enriched (*Supplementary file 1*). Immunostaining with a cross-reactive anti-Sox2 antibody revealed Sox2 expression in developing ampullary organs as well as neuromasts (*Figure 2E–H*), consistent with *Sox2* mRNA expression by in situ hybridization (not shown). Sox2 represses *Atoh1* in supporting cells in the mouse cochlea while, conversely, Atoh1 represses *Sox2* in developing hair cells (*Dabdoub et al., 2008*). By stage 46, paddlefish *Atoh1* expression is hard to detect in canal neuromasts - perhaps partly due to strong expression in scattered overlying epidermal cells, presumably Merkel cells (*Maricich et al., 2009*; *Whitear, 1989*) - but is still strong in ampullary organs (*Figure 2I*). Sox2 immunoreactivity remains strong in both neuromasts and ampullary organs at stage 46 (*Figure 2J*), where it is most likely restricted to supporting cells, given its punctate staining pattern.

Of the 66 genes in the lateral line-enriched dataset that encode known transcription factors (three other transcription factor genes in the dataset were assigned only to families) (*Supplementary file 1*), our top candidate was *Pou4f3* (*Brn3c*, *DFNA15*), which is 27.5-fold lateral line-enriched (*Supplementary file 1*). *Pou4f3* is a confirmed Atoh1 target in hair cells (*Masuda et al., 2011*; *Ikeda et al., 2015*) and required for hearing (*Costaridis et al., 1996*). In mouse cochlear explants, Six1 and Eya1 - acting independently of Atoh1 - are sufficient to induce *Pou4f3*, which promotes hair cell differentiation (*Ahmed et al., 2012*). Like *Atoh1*, paddlefish *Pou4f3* is expressed in ampullary organs, as well as neuromasts (*Figure 2K,L*).

All three *SoxB1* subfamily genes are putative Atoh1 targets in the postnatal mouse cerebellum (*Klisch et al., 2011*) and present in the lateral line-enriched dataset. *Sox3* is 5.2-fold lateral line-enriched (*Supplementary file 1*); as noted above, we previously reported its expression in paddlefish ampullary organs and neuromasts (*Modrell et al., 2011b*). *Sox1* is 8.6-fold lateral line-enriched (*Supplementary file 1*) and expressed in ampullary organs from their eruption (*Figure 2M–P*). It is transiently expressed in the migrating posterior lateral line primordium (*Figure 2M,N*) and, at later stages, expression is also seen in neuromast canal lines (*Figure 2O,P*).

Taken together, these data support a high degree of conservation in the transcriptional network regulating the development of both lateral line organ types, including the likely requirement of Atoh1 for electroreceptor as well as hair cell formation.

## The proneural transcription factor gene Neurod4 is expressed in ampullary organs but not neuromasts

Another highly lateral line-enriched transcription factor gene is *Neurod4 (Ath3, NeuroM)* (18.9-fold enriched; *Supplementary file 1*), a putative cerebellar Atoh1 target (*Klisch et al., 2011*). In mice, this *atonal*-related proneural bHLH transcription factor gene is expressed in the brain, spinal cord, retina, trigeminal ganglia and dorsal root ganglia (*Takebayashi et al., 1997*). It is required for the survival of cerebellar granule cell precursors (*Tomita et al., 2000*), and cooperates with other bHLH and/or homeodomain transcription factors to regulate the development of retinal bipolar and amacrine cells (see *Hatakeyama and Kageyama, 2004*), and trigeminal and facial branchiomotor

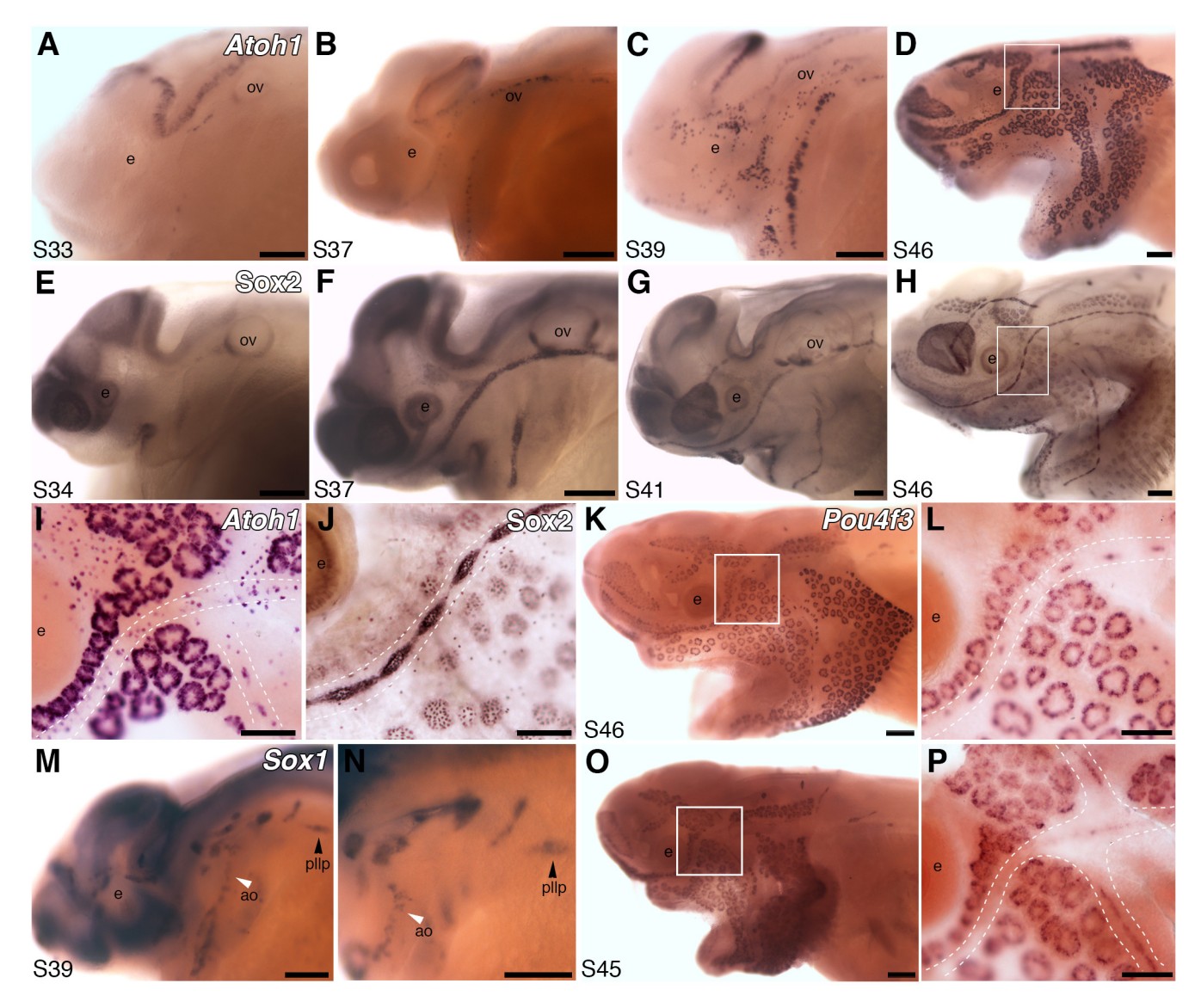

**Figure 2.** Both ampullary organs and neuromasts express transcription factor genes essential for hair cell development, and selected putative Atoh1 targets. (A–D) In situ hybridization for paddlefish *Atoh1* at stages 33, 37, 39 and 46, respectively. (E–H) Sox2 immunostaining at stages 34, 37, 41 and 46, respectively. (I,J) Higher-power views of stage 46 skin-mounts of *Atoh1* (I) and Sox2 (J). Dotted lines indicate approximate boundaries of neuromast containing lateral line canal lines. (K,L) In situ hybridization for *Pou4f3* at stage 46 reveals expression in both ampullary organs and neuromasts. (M,N) In situ hybridization for *Sox1* at stage 39 shows expression in the posterior lateral line primordium (pllp) and ampullary organs erupting on the operculum. (O,P) At later stages, *Sox1* is maintained in ampullary organs and expressed in neuromast canal lines, as shown here at stage 45. Abbreviations: ao, ampullary organ; e, eye; nm, neuromast; ov, otic vesicle, pllp, posterior lateral line primordium. Scale bars: A-H,K,M-O, 200 μm; I,J,L,P, 100 μm.

neurons (*Ohsawa et al., 2005*). In zebrafish, *Neurod4* is important for olfactory neuron development (*Madelaine et al., 2011*); it is transiently expressed during chicken otic neurogenesis, but not hair cell formation (*Bell et al., 2008*). Although the related gene *Neurod1* seems to be important for zebrafish neuromast hair cell differentiation (*Sarrazin et al., 2006*), we previously showed that paddlefish *Neurod1* is not expressed in lateral line organs, only in cranial sensory ganglia (*Modrell et al., 2011b*). *Neurod4* is the only *Neurod* family member in the lateral line-enriched dataset. At early stages, it is expressed in the brain, olfactory epithelium, eyes and trigeminal ganglion (*Figure 3A*). The earliest lateral line expression of *Neurod4* is in developing ampullary organ fields

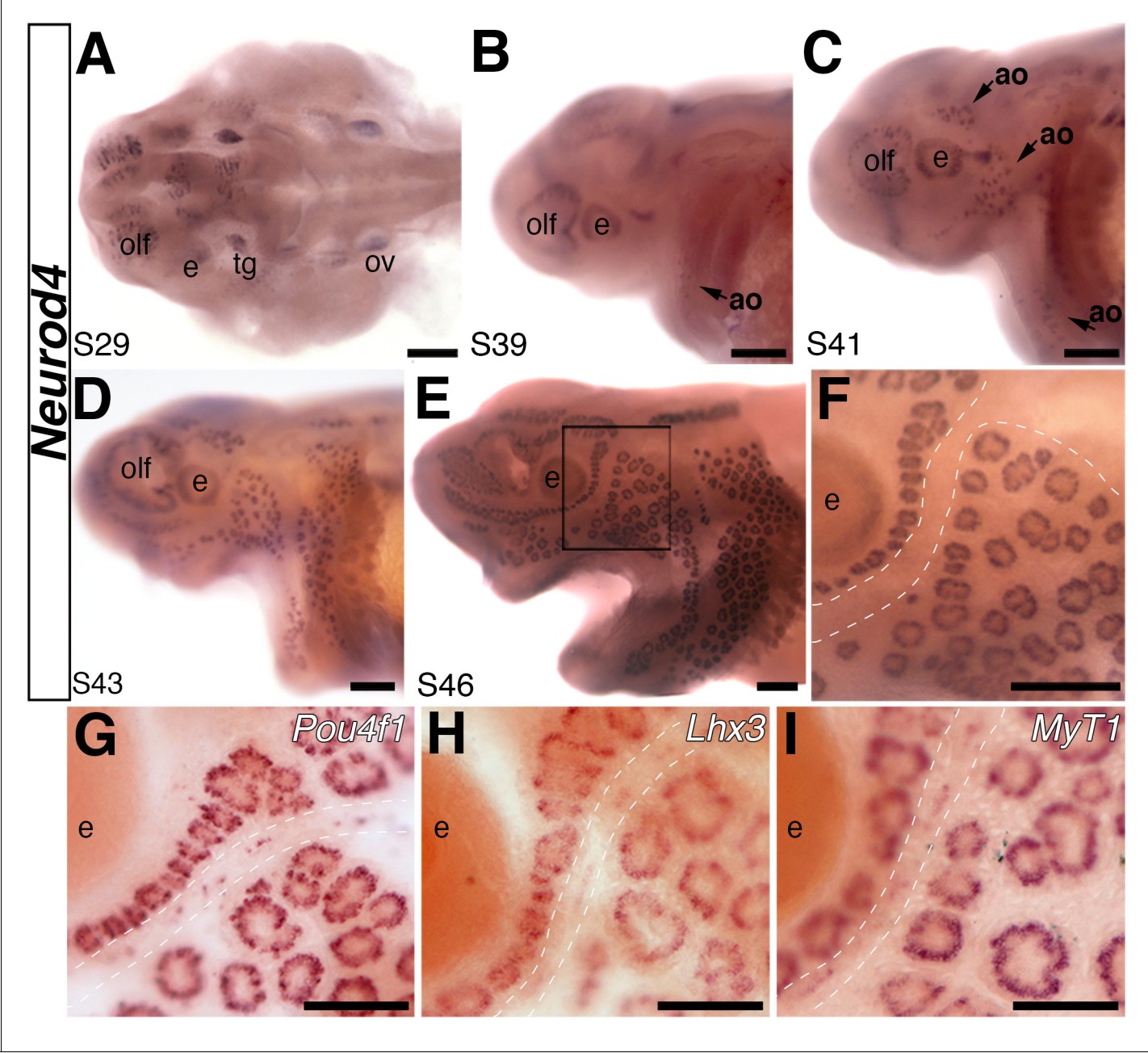

**Figure 3.** The proneural transcription factor gene *Neurod4* is expressed in ampullary organs but not neuromasts. (A–F) In situ hybridization for *Neurod4* at stages 29 (A), 39 (B), 41 (C) 43 (D) and 46 (E). At the earliest stages, transcripts are observed in the olfactory epithelium, eye, trigeminal ganglion. By stage 39 transcripts are observed in the developing ampullary organ fields of the operculum, rapidly expanding to other ampullary organ fields in older embryos. Expression is limited to ampullary organs and is not observed at any stage in neuromasts, as clearly seen at higher power at stage 46 (F). Dotted lines indicate approximate boundaries of neuromast canal lines. (G–I) Potential *Neurod4* interactors *Pou4f1* (G), *Lhx3* (H) and *Myt1* (I) are expressed in both ampullary organs and neuromasts. Abbreviations: ao, ampullary organ; e, eye; nm, neuromast; olf, olfactory epithelium; ov, otic vesicle; tg, trigeminal ganglion. Scale bars: A-F, 200 µm; G-I, 100 µm.

around stage 39 (*Figure 3B*), where it persists (*Figure 3C–F*). Expression is never observed in developing neuromast lines. No other gene has previously been reported to show differential expression in ampullary organs and neuromasts. Hence, *Neurod4* may be important for specifying ampullary organ and/or electroreceptor fate.

We went on to examine the expression of transcription factor genes with known links to *Neurod4* in other cell types. In trigeminal neurons, Pou4f1 (Brn3a) directly represses *Neurod4* (*Lanier et al., 2007*). *Pou4f1* is a putative Atoh1 target in the postnatal mouse cerebellum (*Klisch et al., 2011*), and is 20.6-fold lateral line-enriched (*Supplementary file 1*). Expression was seen in both ampullary organs and neuromasts (*Figure 3G*), suggesting that additional factors besides Pou4f1 are likely involved in controlling the differential expression of *Neurod4* in ampullary organs versus neuromasts.

In the spinal cord, Neurod4 cooperatively interacts with the LIM homeodomain transcription factor Lhx3 to specify motor neurons (*Lee and Pfaff, 2003*). Lhx3 is expressed in all inner ear hair cells, although its expression is differentially regulated by Pou4f3 in cochlear versus vestibular hair cells (*Hertzano et al., 2007*); it is also a putative cerebellar Atoh1 target gene (*Klisch et al., 2011*). *Lhx3* is 16.3-fold lateral line-enriched (*Supplementary file 1*), and proved to be expressed in both ampullary organs and neuromasts (*Figure 3H*). Hence, it is possible that Lhx3 could interact with Neurod4 in paddlefish ampullary organs to specify electroreceptors, and with another partner in neuromasts to specify hair cells.

Finally, we examined the zinc finger transcription factor gene *Myt1*, another putative cerebellar Atoh1 target gene (*Klisch et al., 2011*) that is 6.1-fold lateral line-enriched (*Supplementary file 1*). We selected *Myt1* because its expression is upregulated in *Xenopus* embryos by Neurod4 (*Perron et al., 1999*; *Hardwick and Philpott, 2015*), and it synergizes with Neurod4 to induce neuronal differentiation in this system (*Perron et al., 1999*). Furthermore, *Myt1* transcripts are reportedly enriched in both cochlear and vestibular hair cells in postnatal mice (*Elkon et al., 2015*). We found expression of paddlefish *Myt1* in both ampullary organs and neuromasts (*Figure 3I*).

Overall, our results suggest that the transcription factor networks underlying hair cell development, which center around Atoh1, are likely to be active in paddlefish ampullary organs as well as neuromasts. This suggests significant conservation between the molecular mechanisms underlying hair cell and electroreceptor development. Furthermore, our unbiased RNA-seq approach has also identified the first transcription factor gene expressed in ampullary organs but not neuromasts, *Neurod4*. Given the importance of members of the proneural bHLH Neurod family for specifying cell fate, we suggest that Neurod4 may be involved in specifying ampullary organ and/or electroreceptor fate in paddlefish.

## Hair cell ribbon synapse genes are also expressed in ampullary organs

Some of the most highly lateral line-enriched genes in our paddlefish dataset are required for synaptic transmission in hair cells, which occurs at specialized 'ribbon synapses' characterized by electron-dense, pre-synaptic structures called 'synaptic ribbons' (*Matthews and Fuchs, 2010*; *Pangršič et al., 2012*; *Safieddine et al., 2012*; *Nicolson, 2015*; *Wichmann and Moser, 2015*; *Moser and Starr, 2016*). These tether glutamate-filled synaptic vesicles and stabilize L-type voltage-gated calcium channels at the plasma membrane ($Ca_v1.3$ in hair cells; $Ca_v1.4$, in retinal photoreceptors; *Joiner and Lee, 2015*), enabling rapid and sustained glutamate release in response to activation of these calcium channels by membrane depolarization (*Matthews and Fuchs, 2010*; *Pangršič et al., 2012*; *Safieddine et al., 2012*; *Nicolson, 2015*; *Wichmann and Moser, 2015*; *Moser and Starr, 2016*). Electroreceptors also have synaptic ribbons of varying morphology (*Northcutt, 1986*; *Bodznick and Montgomery, 2005*): in paddlefish, they were described as synaptic 'sheets' (*Jorgensen et al., 1972*). In dissected skate ampullary organ preparations, activation of L-type voltage-gated calcium channels in the basal electroreceptor membrane results in release of a 'glutamate-like' neurotransmitter (*Bennett and Obara, 1986*).

In hair cells, glutamate is loaded into synaptic vesicles by the vesicular glutamate transporter Vglut3, which is encoded by *Slc17a8* (*DFNA25*) and essential for hair cell synaptic transmission in mouse (*Ruel et al., 2008*; *Seal et al., 2008*) and zebrafish (*Obholzer et al., 2008*). This is unusual: Vglut1 or Vglut2 are used at glutamatergic synapses in the central nervous system, and at photoreceptor and bipolar cell ribbon synapses (see e.g. *Zanazzi and Matthews, 2009*; *Pangršič et al., 2012*). *Slc17a8* is one of the most highly enriched genes in our paddlefish dataset (30.9-fold enriched; *Supplementary file 1*), and proved to be expressed in both ampullary organs and neuromasts (*Figure 4A*). Hence, synaptic vesicles in electroreceptors are likely to be loaded by the same vesicular glutamate transporter as hair cells. Furthermore, this also provides independent evidence that the afferent neurotransmitter released by non-teleost electroreceptors is indeed glutamate, as

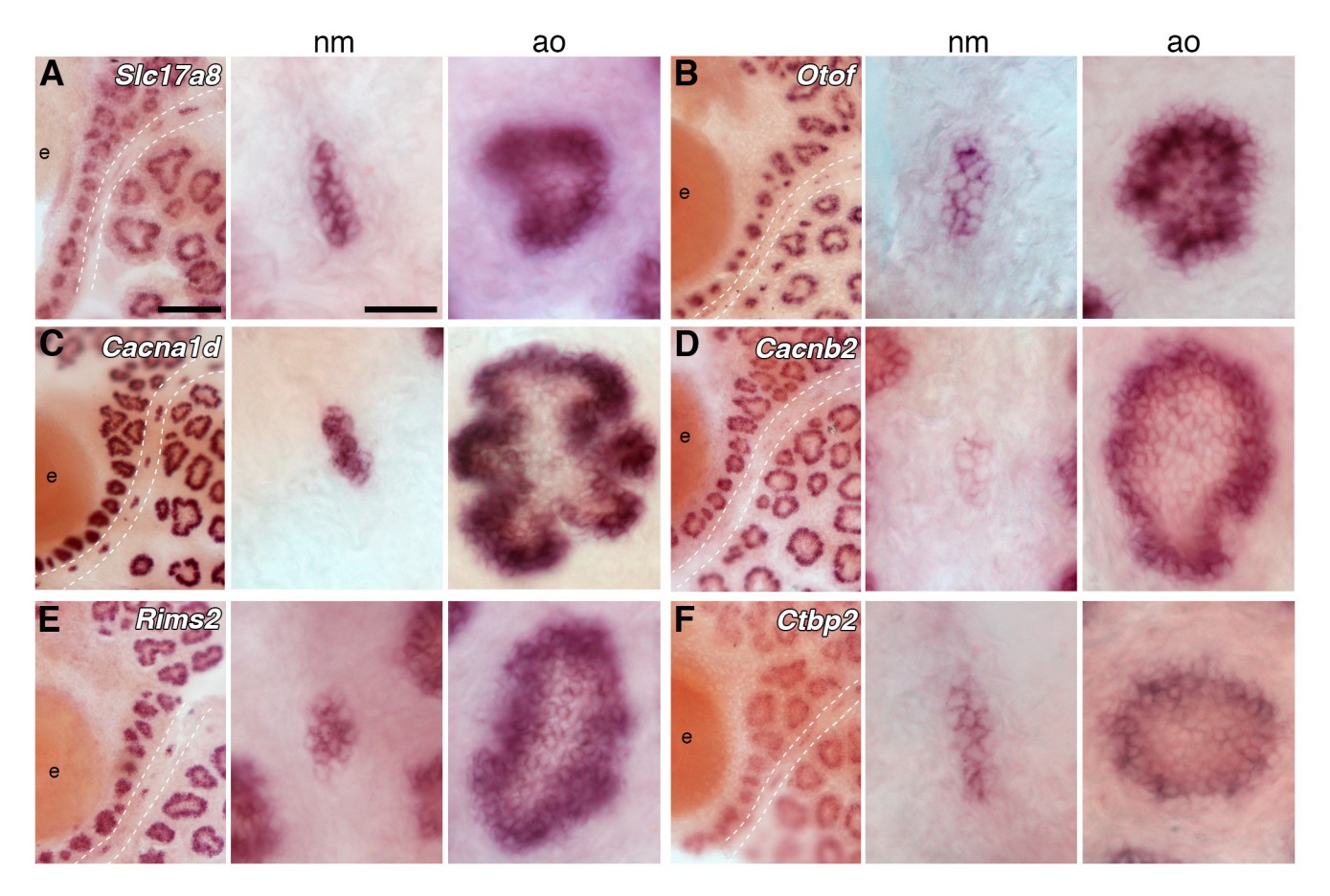

**Figure 4.** Ampullary organs express genes required for transmission at the hair cell ribbon synapse. In situ hybridization at stage 46 reveals expression in both ampullary organs and neuromasts of: (**A**) *Slc17a8*, encoding the vesicular glutamate transporter 3 (Vglut3); (**B**) *Otof*, encoding otoferlin; (**C**) *Cacna1d*, encoding the pore-forming alpha subunit of $Ca_v1.3$; (**D**) *Cacnb2*, encoding an auxiliary beta subunit that is associated with $Ca_v1.3$ in hair cells - note that the level of *Cacnb2* in neuromasts is weaker than in ampullary organs; (**E**) *Rims2*, associated with synaptic ribbons in photoreceptors and hair cells; (**F**) the Ribeye-specific A domain of *Ctbp2*, encoding the ribbon-specific protein Ribeye. Scale bars: 100 µm and 20 µm.

in hair cells (and photoreceptors), as suggested by electrophysiology experiments on dissected skate ampullary organs (*Bennett and Obara, 1986*).

Hair cells are thought to be unique in depending on the multi-$C_2$ domain transmembrane protein otoferlin for synaptic vesicle exocytosis (*Yasunaga et al., 1999*; *Roux et al., 2006*; *Pangršič et al., 2010*; *Chatterjee et al., 2015*; *Strenzke et al., 2016*; *Vogl et al., 2016*), rather than neuronal SNAREs (*Nouvian et al., 2011*). This contrasts not only with conventional synapses but also with all other ribbon synapses (see e.g. *Zanazzi and Matthews, 2009*; *Mercer and Thoreson, 2011*). Oto-ferlin is a type II ferlin (*Lek et al., 2012*) encoded by *Otof* (*DFNAB6*, *DFNAB9*), which is 20.9-fold lateral line-enriched (*Supplementary file 1*). *Otof* is expressed in both ampullary organs and neuromasts in paddlefish (*Figure 4B*). This suggests that synaptic vesicle exocytosis at the electroreceptor ribbon synapse, just as at the hair cell ribbon synapse, is otoferlin-dependent.

The L-type voltage-gated calcium channel whose opening triggers synaptic vesicle exocytosis in hair cells is $Ca_v1.3$ (*Kollmar et al., 1997*; *Platzer et al., 2000*; *Brandt et al., 2003*; *Michna et al., 2003*; *Dou et al., 2004*; *Brandt et al., 2005*; *Baig et al., 2011*). This contrasts with retinal photoreceptors, which do express $Ca_v1.3$, but rely on $Ca_v1.4$ for calcium influx (*Matthews and Fuchs, 2010*; *Joiner and Lee, 2015*). The pore-forming (alpha) subunit of $Ca_v1.3$ is encoded by *Cacna1d*, which is required for hearing (*Platzer et al., 2000*; *Dou et al., 2004*; *Baig et al., 2011*), and also for hair cell function in zebrafish (*Nicolson et al., 1998*; *Sidi et al., 2004*), where it is expressed in both

neuromasts and the inner ear (*Sidi et al., 2004*). (The two zebrafish *cacna1d* genes show differential expression: *cacna1da* is expressed in hair cells plus retinal photoreceptors, while *cacna1db* is expressed in retinal and pineal photoreceptors, but not hair cells; *Sidi et al., 2004*.) Paddlefish *Cacna1d* is 19.3-fold lateral line-enriched (*Supplementary file 1*) and expressed in both ampullary organs and neuromasts (*Figure 4C*). Hence, $Ca_v1.3$ channels in the basal membrane are likely to mediate glutamate release from both paddlefish electroreceptors and hair cells. Given the homology of non-teleost ampullary organs (*Bullock et al., 1983*; *Northcutt, 1986*, *Northcutt, 1992*; *Braun, 1996*; *New, 1997*; *Baker et al., 2013*), this also suggests that the L-type voltage-gated calcium channels involved in neurotransmitter release from skate ampullary organs are likely to be $Ca_v1.3$ channels.

The abundance and function of $Ca_v1.3$ channels in inner ear hair cells is regulated by the auxiliary beta subunit $Ca_v\beta_2$, which is required for hearing (*Neef et al., 2009*). $Ca_v\beta_2$ is encoded by *Cacnb2*, which is the only other voltage-gated $Ca^{2+}$ channel subunit gene in the lateral line-enriched dataset (2.7-fold enriched; *Supplementary file 1*). Although neuromast expression of *cacnb2a* and *cacnb2b* was not reported in zebrafish (*Zhou et al., 2008*), paddlefish *Cacnb2* is expressed in both ampullary organs and neuromasts, with seemingly stronger expression in ampullary organs (*Figure 4D*). This suggests that $Ca_v\beta_2$ may be the auxiliary beta-subunit for $Ca_v1.3$ channels in electroreceptors, as well as hair cells.

Furthermore, given that the voltage sensors in skate electroreceptors are L-type voltage-gated calcium channels in the apical membrane (*Bennett and Obara, 1986*; *Lu and Fishman, 1995*; *Bodznick and Montgomery, 2005*), recently demonstrated to be $Ca_v1.3$ channels (*Bellono et al., 2017*), and the fact that an apical calcium conductance is required for the firing of sturgeon ampullary organ afferents (*Teeter et al., 1980*), it seems likely that apically-located $Ca_v1.3$ channels, with $Ca_v\beta_2$ auxiliary beta subunits, act as the voltage-sensing channels, as well as mediating neurotransmitter release basally.

Finally, we report the expression in both ampullary organs and neuromasts of *Rims2* (*Rim2*) and *Ctbp2* (*Ribeye*), which encode proteins associated with synaptic ribbons in photoreceptors as well as hair cells (*Matthews and Fuchs, 2010*; *Pangršič et al., 2012*; *Safieddine et al., 2012*; *Nicolson, 2015*; *Wichmann and Moser, 2015*) (*Figure 4E,F*). *Rims2* encodes Rab3-interacting molecules 2α and β, which are required in cochlear inner hair cells for $Ca_v1.3$ channel recruitment to the active zone membrane beneath the synaptic ribbon (*Jung et al., 2015*). *Rims2* is 15.7-fold lateral line-enriched (*Supplementary file 1*) and expressed in all lateral line organs (*Figure 4E*). Hence, Rims2 may also be involved in $Ca_v1.3$ channel recruitment in electroreceptors. Ribeye, the only known synaptic ribbon-specific protein, is the main structural component of synaptic ribbons in both photoreceptors and hair cells, encoded by usage of an alternative start site for the transcription factor gene *Ctbp2* that generates an N-terminal A-domain unique to Ribeye (*Matthews and Fuchs, 2010*; *Nicolson, 2015*; *Wichmann and Moser, 2015*). Ribeye is important in zebrafish neuromast hair cells for $Ca_v1.3$ channel recruitment to synaptic ribbons and stabilizing synaptic contacts with afferent neurons (*Sheets et al., 2011*; *Lv et al., 2016*). Recently, deletion in mice of the exon encoding the A-domain showed that Ribeye is essential in the retina (the ear was not examined) both for ribbon formation per se, and for rapid and sustained neurotransmitter release (*Maxeiner et al., 2016*). *Ctbp2* is not in the lateral line-enriched dataset, but was present in the combined transcriptome, so was easily cloned. As expected, a riboprobe that exclusively recognizes the Ribeye-specific A-domain sequence of paddlefish *Ctbp2* reveals expression in both ampullary organs and neuromasts (*Figure 4F*), suggesting that Ribeye is likely to be a key component of synaptic ribbons in electroreceptors (and hair cells), where it may also be important for $Ca_v1.3$ channel recruitment.

Taken together, these data suggest that the mechanisms of neurotransmission at the ribbon synapse in paddlefish electroreceptors are essentially identical to those at the hair cell ribbon synapse, involving otoferlin-dependent exocytosis of synaptic vesicles, loaded with glutamate by Vglut3, in response to the activation of basal $Ca_v1.3$ channels. Since non-teleost ampullary organs are homologous (*Bullock et al., 1983*; *Northcutt, 1986*, *Northcutt, 1992*; *Braun, 1996*; *New, 1997*; *Baker et al., 2013*), we predict that these mechanisms will be conserved across all other non-teleost ampullary organs.

## Differential expression of beta-parvalbumin genes in ampullary organs and neuromasts

The paddlefish lateral line-enriched dataset contains two *parvalbumin (Pvalb)* genes: one, annotated as being related to zebrafish '*pvalb8*', is enriched 22.8-fold; the other, annotated as being related to zebrafish '*pvalb3*', is enriched 2.1-fold (*Supplementary file 1*). Parvalbumins are cytosolic EF-hand $Ca^{2+}$-buffering proteins (*Schwaller, 2010*). $Ca^{2+}$ is essential for multiple aspects of hair cell function (*Lenzi and Roberts, 1994*; *Mammano et al., 2007*; *Ceriani and Mammano, 2012*) and different hair cell subtypes are distinguished by different complements of EF-hand $Ca^{2+}$ buffers, including different parvalbumin family members. Mammals have a single alpha-parvalbumin, encoded by *Pvalb*, and a single beta-parvalbumin, oncomodulin, encoded by *Ocm* (*Schwaller, 2010*). Alpha-parvalbumin is restricted to cochlear inner hair cells, while oncomodulin is restricted to cochlear outer hair cells, and is also expressed in vestibular hair cells (*Sakaguchi et al., 1998*; *Yang et al., 2004*; *Simmons et al., 2010*; *Pangršič et al., 2015*; *Tong et al., 2016*). Non-mammalian species have variable numbers of *parvalbumin* genes (nine in zebrafish; *Friedberg, 2005*), and the nomenclature for both genes and proteins is inconsistent and confusing. This led us to undertake a phylogenetic analysis of selected vertebrate parvalbumin proteins, including the two predicted paddlefish proteins (*Figure 5A*). This revealed three distinct clades: one comprising the alpha-parvalbumins (including zebrafish 'Pvalb6' and 'Pvalb7'; *Friedberg, 2005*), and two containing beta-parvalbumins (*Figure 5A*). One of the beta-parvalbumin clades includes the oncomodulins (i.e., mammalian beta-

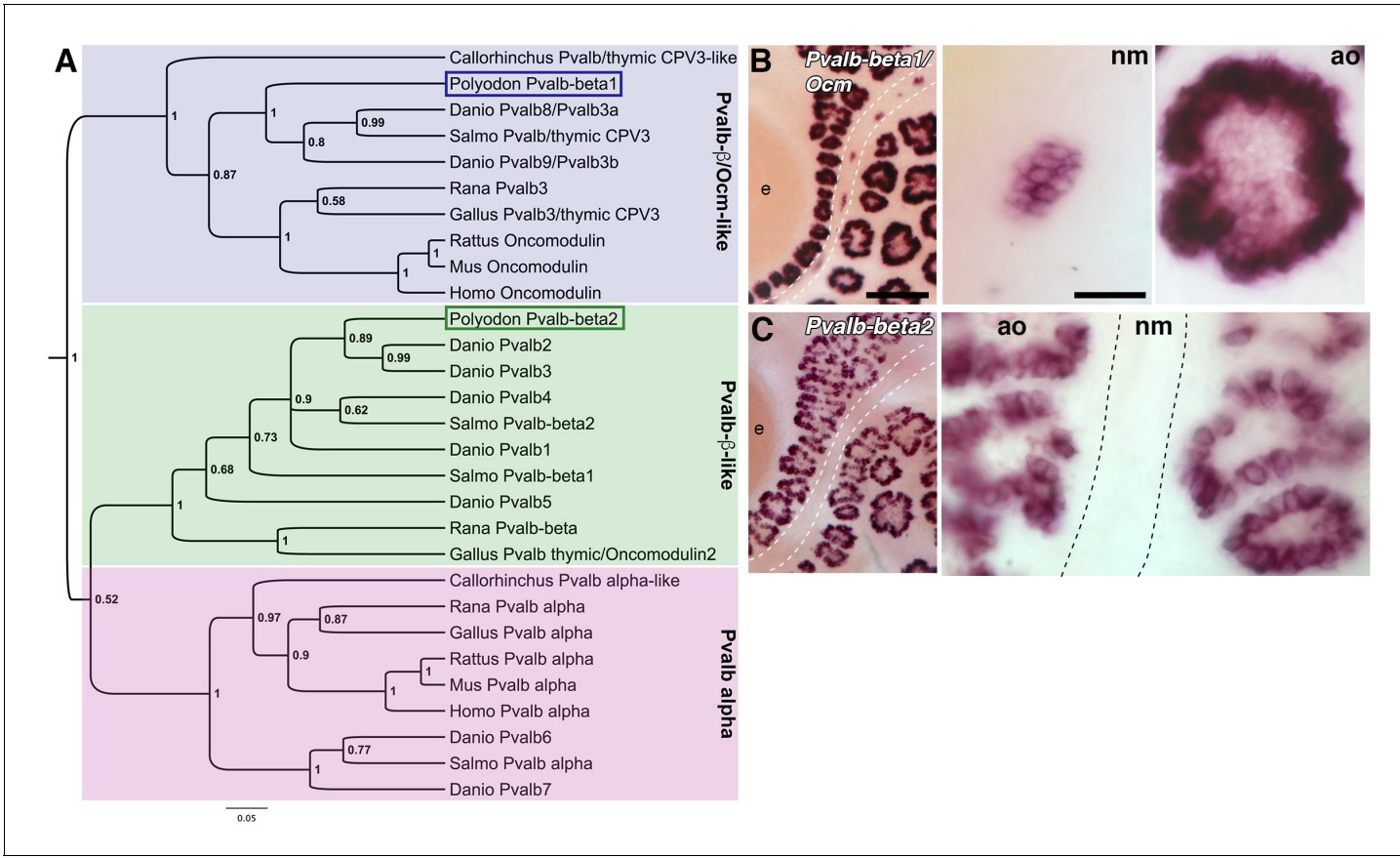

**Figure 5.** Two *beta-parvalbumin* genes are differentially expressed in ampullary organs versus neuromasts. (A) Phylogenetic analysis of selected vertebrate parvalbumin proteins shows three clades: the alpha-parvalbumins, plus two clades containing beta-parvalbumins: the first includes the mammalian beta-parvalbumins, i.e., oncomodulins, and chicken Pvalb3/thymic CPV3. One of the paddlefish lateral line-enriched parvalbumin genes encodes a protein that groups within the beta-parvalbumin clade containing the oncomodulins, so we have named it Pvalbβ1/Ocm. The other falls within the second beta-parvalbumin clade, so we have named it Pvalbβ2. (B) In situ hybridization shows that *Pvalbβ1/Ocm* is expressed in both ampullary organs and neuromasts, while (C) *Pvalbβ2* is restricted to ampullary organs. Scale bars: 100 μm and 20 μm.

parvalbumins), chicken Pvalb3/thymic CPV3 (encoded by *Ocm*) (*Hapak et al., 1994*), bullfrog Pvalb3 (*Heller et al., 2002*) and zebrafish 'Pvalb8' and 'Pvalb9', which were originally named Pvalb3a and Pvalb3b owing to their similarity to chicken Pvalb3/CPV3 and oncomodulin (*Hsiao et al., 2002*; *Friedberg, 2005*) (*Figure 5A*). The second beta-parvalbumin clade includes chicken thymic Pvalb (avian thymic hormone, encoded by *Ocm2*) (*Brewer et al., 1989*), zebrafish 'Pvalb1-5' (*Friedberg, 2005*) and various other beta-parvalbumins (*Figure 5A*).

Following our phylogenetic analysis, it was clear that both lateral line-enriched *Pvalb* genes encode beta-parvalbumins. We named the most highly lateral line-enriched gene *Pvalbβ1/Ocm* (22.8-fold enriched; 'pvalb8' in *Supplementary file 1*), because the predicted protein groups in the beta-parvalbumin clade containing mammalian oncomodulins (*Figure 5A*). *Pvalbβ1/Ocm* is expressed in both ampullary organs and neuromasts (*Figure 5B*). Interestingly, 'Pvalb8' was reported to be the most highly expressed transcript in skate ampullary organs: the authors suggest that, following the voltage-dependent $Ca^{2+}$ influx via $Ca_v1.3$ that depolarizes the electroreceptor and activates BK, this parvalbumin could bind $Ca^{2+}$, thus blocking BK-mediated hyperpolarization and enabling further oscillations (*Bellono et al., 2017*).

We named the less highly lateral line-enriched gene *Pvalbβ2* (2.1-fold enriched; 'pvalb3' in *Supplementary file 1*), as it falls into the second beta-parvalbumin clade (*Figure 5A*). At all stages examined, *Pvalbβ2* proved to be expressed in ampullary organs but not neuromasts (*Figure 5C*). Therefore, in addition to *Neurod4*, we have identified another transcript expressed by ampullary organs but not neuromasts. We suggest that this beta-parvalbumin is likely to be involved in ampullary organ-specific aspects of $Ca^{2+}$ regulation.

## Identification of ampullary organ-specific voltage-gated potassium channel subunits

The oscillatory character of the basal membrane voltage of skate electroreceptors depends on voltage-gated potassium channels, which contribute to repolarization of the basal membrane (*Bennett and Obara, 1986*; *Lu and Fishman, 1995*; *Bodznick and Montgomery, 2005*). Noisy voltage oscillations have been recorded from ampullary organ canals in adult paddlefish in vivo (*Neiman and Russell, 2004*). Hence, we were very interested to find two highly lateral line-enriched *shaker*-related voltage-gated potassium channel subunit genes in our paddlefish dataset: *Kcna5*, encoding the pore-forming alpha subunit of $K_v1.5$ (12.4-fold enriched) and *Kcnab3*, encoding the beta subunit $K_vβ3$ (21.4-fold enriched). In contrast to the L-type voltage-gated calcium channel subunit genes *Cacna1d* and *Cacnb2*, which are expressed in both ampullary organs and neuromasts (*Figure 4C,D*), *Kcna5* (*Figure 6A–D*) and *Kcnab3* (*Figure 6E–H*) are only expressed in ampullary organs. Expression of both genes is first seen around stages 38–39, slightly after the eruption of the first ampullary organs at the surface at stage 37 (*Modrell et al., 2011a*).

Voltage-gated potassium channels are tetramers of four alpha (pore-forming) subunits, each containing six transmembrane segments, of which S4 contains a high density of positively charged residues and is the main transmembrane voltage-sensing component (*Barros et al., 2012*). Intriguingly, paddlefish $K_v1.5$ has a number of amino acid substitutions at otherwise highly conserved positions in S4, as well as in the S3-S4 linker, at the top of the S5 helix, and in the channel pore (*Figure 6I,J*). How the sum of these substitutions affects paddlefish $K_v1.5$ channel behavior must await studies of channel expression.

## Discussion

Here, we took advantage of the abundance of ampullary organs (plus some neuromasts) on the operculum (gill-flap) of late-larval Mississippi paddlefish, and the absence of lateral line organs on the fins (*Modrell et al., 2011a*, *2011b*), to generate a dataset of over 400 identified genes whose transcripts are enriched at least 1.8-fold in operculum versus fin tissue (i.e., lateral line-enriched). The dataset is not exhaustive: it does not include some genes whose expression we have validated in developing ampullary organs and neuromasts in paddlefish, whether via previous candidate gene approaches (e.g. *Eya* family members and *Six* family members other than *Six1* and *Six2*; *Modrell et al., 2011a*), or some genes cloned in this study from transcripts present in the combined operculum plus fin transcriptome (e.g. *Ctpb2*, encoding the synaptic ribbon protein Ribeye). Nevertheless, this unbiased dataset provides an important foundation for investigating the molecular basis

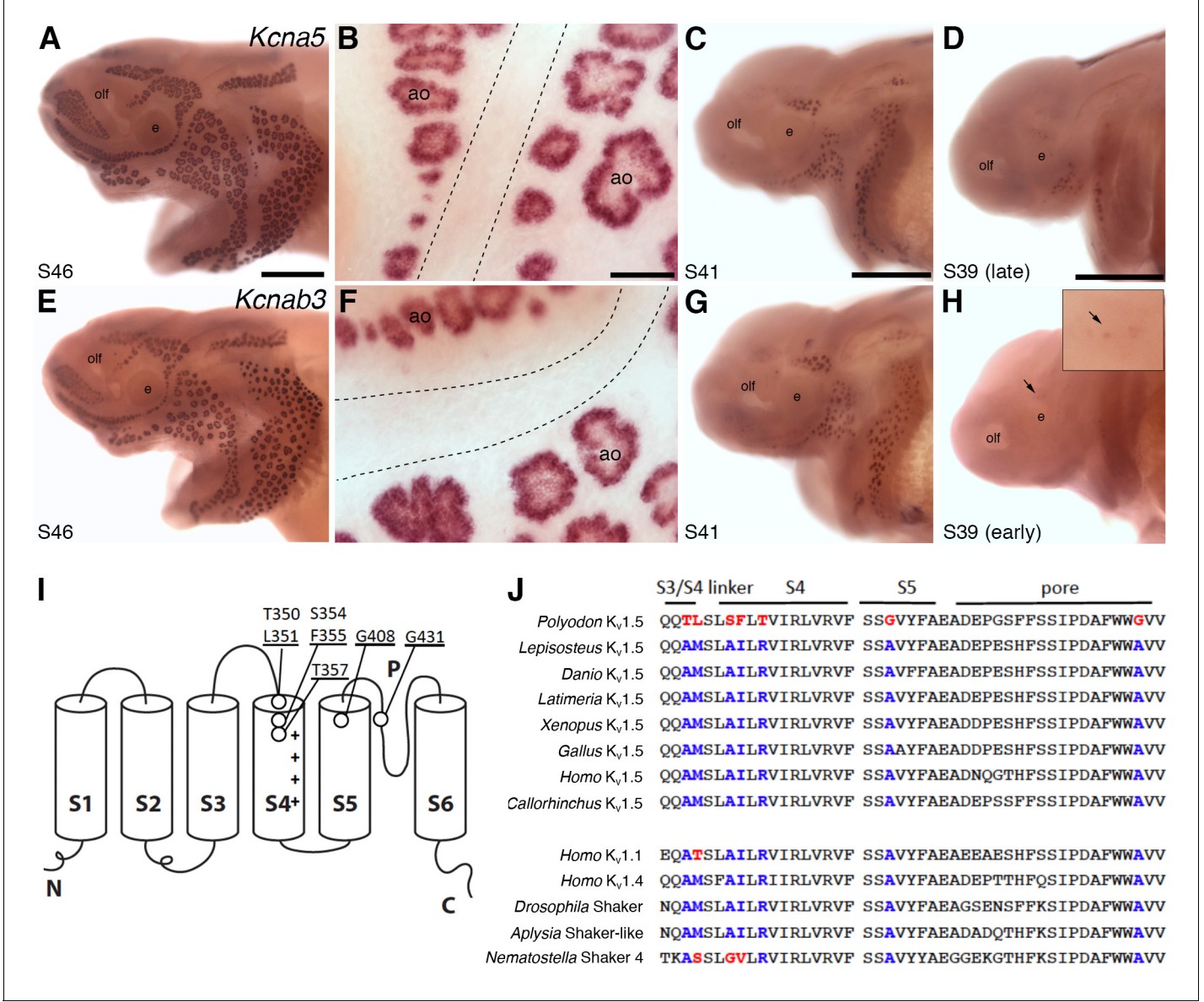

**Figure 6.** *Shaker*-related voltage-gated potassium channel subunit genes expressed in ampullary organs but not neuromasts. (A,B) In situ hybridization at stage 46 for *Kcna5*, which encodes the pore-forming alpha subunit of the voltage-gated potassium channel $K_v1.5$, reveals expression only in the developing ampullary organ fields. Dotted lines indicate approximate boundaries of neuromast canal lines. (C,D) Even at the successively earlier stages shown (stage 41 and 39), *Kcna5* is still restricted to the developing ampullary organ fields. (E–H) Expression of *Kcnab3*, encoding the auxiliary beta subunit $K_v\beta3$, is similarly confined to the developing ampullary organ fields. Dotted lines indicate approximate boundaries of neuromast canal lines. The arrow in H, and the higher-power view of this region shown in the inset, indicate the area where the first *Kcnab3* expression is noted, at stage 39. Scale bars: A,C-E,G,H, 0.5 mm; B,F, 50 μm. Abbreviations: ao, ampullary organ; e, eye; olf, olfactory pit. (I) Schematic, linear structure of the pore-forming alpha subunit of a $K_v$ channel, with the positions noted of amino acid substitutions in paddlefish $K_v1.5$. (J) Amino acid sequences across the S3/4 linker region, the voltage-sensing segment S4, plus S5 and the pore, from paddlefish $K_v1.5$ (top) and other Shaker-related $K_v$ channels for comparison, indicating the deep conservation of some of these amino acid positions across metazoans. The first set of sequences are from $K_v1.5$ across the jawed vertebrates, including three ray-finned bony fishes: *Polyodon spathula* (Mississippi paddlefish, a non-teleost chondrostean fish), *Lepisosteus oculatus* (spotted gar, a non-teleost neopterygian fish), *Danio rerio* (zebrafish, a teleost neopterygian fish); four lobe-finned fishes/tetrapods: *Latimeria chalumnae* (coelacanth), *Xenopus tropicalis* (tropical clawed frog), *Gallus gallus* (chicken), *Homo sapiens* (human); and a cartilaginous fish (*Callorhinchus milii*, a holocephalan). The second set of sequences are from two other human Shaker-related channels ($K_v1.1$ and $K_v1.4$), and three invertebrate Shaker orthologs, from *Drosophila melanogaster* (an insect, i.e., an ecdysozoan), *Aplysia californica* (a mollusc, i.e., a spiralian) and *Nematostella vectensis* (a sea anemone, i.e., a cnidarian).

of ampullary organ development. Validation of a selection of genes from the dataset revealed significant molecular conservation between developing paddlefish ampullary organs and neuromasts, both in their expression of transcription factor genes critical for hair cell development, and also of genes essential for transmission specifically at the hair cell ribbon synapse. For the first time in any vertebrate, we also identify genes expressed in ampullary organs but not neuromasts, including a transcription factor, a beta-parvalbumin and voltage-gated potassium channel subunits consistent with predictions from skate ampullary organ electrophysiology.

## Atoh1 and Neurod4 are likely to be critical for ampullary organ/ electroreceptor development

Key elements of the transcription factor network underlying hair cell development, centering on the bHLH transcription factor Atoh1 (*Cai and Groves, 2015*; *Jahan et al., 2015*; *Costa et al., 2017*), seem to be conserved in developing paddlefish ampullary organs, including expression of *Six1*, *Eya1* (*Modrell et al., 2011a*), Sox2, *Atoh1* itself (this study and *Butts et al., 2014*) and *Pou4f3* (*Brn3c*). Hence, Atoh1 and Pou4f3 are likely to be critical for the specification and differentiation of electroreceptors, as well as hair cells. This further highlights the importance of developmental context for Atoh1 activity, since it is also required for the specification of mechanosensory Merkel cells and proprioceptive neurons, cerebellar granule cells and intestinal secretory cells (*Cai and Groves, 2015*; *Jahan et al., 2015*; *Costa et al., 2017*). It also raises the question of which transcription factors are involved in the differentiation of ampullary organs/electroreceptors versus neuromasts/hair cells, whether acting downstream of Atoh1, or in parallel with it.

Of the 16 transcription factor genes whose expression we have reported to date in developing paddlefish lateral line organs, all except one are expressed in both ampullary organs and neuromasts: these are *Six1*, *Six2*, *Six4*, *Eya1*, *Eya2*, *Eya3*, *Eya4* (*Modrell et al., 2011a*); the three *SoxB1* class genes *Sox1*, Sox2 (this study) and *Sox3* (*Modrell et al., 2011b*); *Atoh1* (this study and *Butts et al., 2014*), Pou4f3, Pou4f1, Lhx3 and Myt1 (this study). The single exception is the proneural bHLH transcription factor gene *Neurod4*, a putative cerebellar Atoh1 target (*Klisch et al., 2011*) present in the lateral line-enriched dataset, which proved to be expressed in developing ampullary organs but not neuromasts. The related family member *Neurod1* (which is expressed in paddlefish cranial sensory ganglia, but not lateral line organs; *Modrell et al., 2011b*) suppresses a hair cell fate in mouse otic neurons, and is important for the specification of outer versus inner hair cells in the cochlea (*Jahan et al., 2010*), while the even more closely related *Neurod6* was identified as enriched in cochlear but not vestibular hair cells (*Elkon et al., 2015*). Hence, different NeuroD family members may be involved in specifying different hair cell subtypes. Taken together, we propose that Neurod4 is likely to be critical in paddlefish for the specification of ampullary organs/electroreceptors within lateral line placode-derived sensory ridges.

The function of these transcription factors during lateral line development could in principle be tested using the RNA-guided nuclease system CRISPR/Cas9, which has successfully been used to generate biallelic mutations efficiently in, for example, F0-injected axolotl (*Fei et al., 2014*; *Flowers et al., 2014*) and lamprey (*Square et al., 2015*). However, the very restricted annual spawning season for paddlefish, and the availability of 1-cell-stage embryos for injection only in a commercial fishery, rather than a laboratory setting, raise significant technical and logistical obstacles to optimizing CRISPR/Cas9 for this particular species. Another non-teleost chondrostean fish, the sturgeon *Acipenser ruthenus* (sterlet), has a much longer spawning season, with 1-cell-stage embryos readily available for microinjection in research facilities (e.g. *Saito et al., 2014*). Therefore, we plan to purse functional experiments on electroreceptor development using this species in the future.

## Electroreceptor synaptic transmission mechanisms are conserved with hair cells

Although it was clear from electron microscopy that non-teleost electroreceptors in all species examined have ribbon-type synapses, given the presence of electron-dense presynaptic bodies ('bars', 'sheets' or 'spheres', depending on the species), surrounded by synaptic vesicles (*Jørgensen, 2005*), nothing was known at the molecular level about transmission mechanisms at electroreceptor synapses, except that activation of L-type voltage-gated calcium channels results in the release of a 'glutamate-like' neurotransmitter (*Bennett and Obara, 1986*). Our data suggest that the electroreceptor

ribbon synapse is glutamatergic, and functions in the same way as the hair cell ribbon synapse (*Safieddine et al., 2012*; *Nicolson, 2015*; *Wichmann and Moser, 2015*; *Moser and Starr, 2016*), with glutamate being loaded into synaptic vesicles by Vglut3, and otoferlin-dependent exocytosis being triggered by the activation of $Ca_v1.3$ channels, potentially including the auxiliary beta subunit $Ca_v\beta_2$, as in cochlear inner hair cells (*Neef et al., 2009*). In contrast, retinal and pineal photoreceptors express Vglut1 and Vglut2 and neuronal SNAREs, and retinal photoreceptors depend on $Ca_v1.4$ (see e.g. *Zanazzi and Matthews, 2009*; *Matthews and Fuchs, 2010*; *Mercer and Thoreson, 2011*; *Joiner and Lee, 2015*). These physiological similarities between electroreceptors and hair cells are consistent with the expression in developing ampullary organs of key hair cell transcription factor genes, some of which presumably control the expression of ribbon synapse-associated genes in both electroreceptors and hair cells.

## The expression of $Ca_v$ and $K_v$ channel subunit genes in paddlefish ampullary organs is consistent with predictions from skate ampullary organ electrophysiology

Our understanding of non-teleost ampullary organ physiology is primarily based on current- and voltage-clamp studies of epithelial currents in dissected single ampullary organ preparations from skates (cartilaginous fishes) (*Bennett and Obara, 1986*; *Lu and Fishman, 1995*; *Bodznick and Montgomery, 2005*). These revealed that L-type voltage-gated calcium channels in the apical (lumenal, i.e., exterior-facing) electroreceptor membrane open in response to low-frequency cathodal electric fields (i.e., lumen-negative at the apical surface of the electroreceptor, relative to the interior of the animal). $Ca^{2+}$ entry depolarizes the apical membrane and subsequently the basal membrane, which activates L-type voltage-gated calcium channels in the basal membrane. $Ca^{2+}$ entry through these channels further depolarizes the basal membrane, resulting in synaptic vesicle exocytosis and neurotransmitter release. Basal $Ca^{2+}$ entry also activates basal voltage-gated potassium channels and calcium-dependent chloride channels, thus repolarizing the basal membrane and de-activating the basal voltage-gated calcium channels (this gives the basal membrane voltage its oscillatory character). The apical membrane is repolarized after sufficient $Ca^{2+}$ enters through the apical voltage-gated calcium channels to trigger the apical calcium-gated potassium channel BK. This terminates the depolarization and oscillations of the basal membrane. Importantly, the apical L-type voltage-gated calcium channels are partially activated in the absence of stimulus by a bias current across the sensory epithelium (provided by sodium/calcium exchangers and sodium/potassium pumps), so neurotransmitter is released steadily at rest, while the activity of the apical BK channel keeps the electroreceptor near threshold at rest and also during prolonged stimulus. Low-frequency cathodal stimuli accelerate the resting discharge of the afferent nerve, while anodal stimuli decelerate and eventually inhibit discharge (*Bennett and Obara, 1986*; *Lu and Fishman, 1995*; *Bodznick and Montgomery, 2005*). Also in skate, $Ca_v1.3$ has just been identified as the L-type voltage-gated calcium channel that mediates the low-threshold voltage-dependent inward current, and works with BK to mediate electroreceptor membrane oscillations (*Bellono et al., 2017*).

Our data on voltage-gated ion channel expression in larval paddlefish ampullary organs are relevant both for the voltage-sensing calcium channels and for the voltage-gated potassium channel involved in rectifying the basal membrane during oscillations. An apical calcium conductance is required for ampullary organ afferent nerve firing in another non-teleost chondrostean fish, the sturgeon *Scaphirhynchus platorynchus* (*Teeter et al., 1980*). The expression in larval paddlefish ampullary organs of *Cacna1d*, encoding the pore-forming alpha subunit of $Ca_v1.3$ channels, and *Cacnb2*, encoding the auxiliary beta subunit $Ca_v\beta_2$, is consistent with the hypothesis that $Ca_v1.3$ channels are candidates for the apical voltage-sensing L-type voltage-gated calcium channels, as well as for the basal L-type voltage-gated calcium channels whose activation triggers otoferlin-dependent synaptic vesicle exocytosis and glutamate release at the ribbon synapse (see previous section). The identification of $Ca_v1.3$ channels as the voltage-sensor in skate electroreceptors (*Bellono et al., 2017*) further supports this hypothesis. Furthermore, given that *Cacnb2* seems to be expressed at higher levels in ampullary organs than in neuromasts in paddlefish, it is possible that the voltage-sensing channels include the auxiliary beta subunit $Ca_v\beta_2$. (In contrast, deep-sequencing data from skate ampullary organs suggest that $Ca_v\beta_1$, encoded by *Cacnb1*, might be the auxiliary beta subunit in this species; *Bellono et al., 2017*.) Testing these hypotheses will require subcellular localization using paddlefish

Ca$_v$ channel subunit-specific antibodies and/or the optimization of CRISPR/Cas9 in conjunction with larval electroreceptor recordings, in this or related species.

We also identified expression in paddlefish ampullary organs, but not neuromasts, of *Kcna5*, encoding the pore-forming subunit of K$_v$1.5 channels, and *Kcnab3*, encoding an auxiliary beta-subunit. In the cardiac atrium, K$_v$1.5 channels are thought to conduct the 'ultra-rapid delayed rectifier' current (I$_{Kur}$), with relatively slow inactivation properties, which contributes to repolarization (*Schmitt et al., 2014*; *Wettwer and Terlau, 2014*). Furthermore, when human *KCNA5* and *KCNAB3* are heterologously co-expressed in cell culture, they form a novel K$_v$ channel that, upon depolarization, mediates very fast-inactivating (A-type) outward currents (*Leicher et al., 1998*). Hence, it is possible that K$_v$1.5 and K$_v\beta$3 in paddlefish ampullary organs together constitute a rapidly-inactivating voltage-gated potassium channel that conducts an ultra-rapid delayed rectifier current. These properties would plausibly be optimal for rapid oscillation of the basal electroreceptor membrane. Noisy voltage oscillations have been recorded from ampullary organ canals in adult paddlefish in vivo (*Neiman and Russell, 2004*). Importantly, the only evidence until now for the existence of a rectifying K$_v$ channel in non-teleost ampullary organs has been from current- and voltage-clamp approaches in dissected skate (cartilaginous fish) ampullary organs (*Bennett and Obara, 1986*; *Lu and Fishman, 1995*; *Bodznick and Montgomery, 2005*). Our gene expression data suggest that this channel also exists in a chondrostean fish, supporting the hypothesis that non-teleost electroreceptor physiology is conserved (at least across jawed vertebrates), and identify K$_v$1.5 and K$_v\beta$3 as the candidate channel.

## Amino acid substitutions may alter K$_v$1.5 channel properties in paddlefish

The sequence of paddlefish *Kcna5* predicts several amino acid substitutions in highly conserved regions of K$_v$1.5 that could affect its properties. The S4 segment of voltage-gated ion channels is the voltage sensor, and is highly conserved (*Barros et al., 2012*). The first positively-charged amino acid in the voltage sensor is normally arginine: since it is the first arginine in S4, it is referred to as the R1 position. In paddlefish K$_v$1.5, this has been replaced by a threonine (T357). Experimental substitution of methionine at this site in *Drosophila* Shaker (R362M; *Aggarwal and MacKinnon, 1996*) decreases the gating charge on the channel and the slope of the conductance voltage curve (*Aggarwal and MacKinnon, 1996*; *Elliott et al., 2012*). Aside from its role as part of the voltage sensor, R1 forms a seal with hydrophobic residues in S2 that keeps ions from flowing around the S4 segment: substitution of R1 in *Drosophila* Shaker with histidine (R362H) allows protons to leak past S4 (*Starace and Bezanilla, 2004*). Substitution with some other amino acids (alanine, cysteine, serine, valine) allows monovalent cations to leak past S4 when the cell is hyperpolarized (omega current) (*Tombola et al., 2005*). It is not known if a threonine at this position would encourage an omega current, but an omega current could act like a slow depolarizing leak current, possibly contributing to the spontaneous activity of ampullary electroreceptors.

The other amino acid substitutions in paddlefish K$_v$1.5 are at conserved positions in the S3-S4 linker, at the top of the S5 helix, and in the channel pore. A variety of studies indicate that the end of the S3/S4 linker and the top of S4 contact the top of S2, S3 and S5/pore in the closed or open state (*Kanevsky and Aldrich, 1999*; *Elliott et al., 2004*; *Soler-Llavina et al., 2006*; *Henrion et al., 2009*; *Lin et al., 2011*; *Elliott et al., 2012*). Generally speaking, amino acid substitutions at some of these amino acid positions cause a rightward shift in the conductance-voltage curve, meaning that the channel would open at more depolarized voltages. Characterization of the effects on K$_v$1.5 of these various amino acid substitutions will require future paddlefish channel expression in a heterologous system and mutagenesis studies: for example, characterizing the properties of paddlefish K$_v$1.5, as compared with paddlefish K$_v$1.5 in which these amino acids (in different combinations) have been substituted with conserved residues, and/or substituting the paddlefish-specific amino acids into, for example, human K$_v$1.5.

## Insights into electroreceptor evolution

As noted above, of the 16 transcription factor genes whose expression we have reported to date in developing paddlefish lateral line organs (this study; *Modrell et al., 2011a*, *2011b*; *Butts et al., 2014*), all except one (*Neurod4*) are expressed in ampullary organs as well as neuromasts, including

transcription factor genes that are essential for hair cell development, such as *Atoh1* and *Pou4f3* (*Brn3c*). Furthermore, genes that are required for synaptic transmission at the hair cell (but not photoreceptor) ribbon synapse are also expressed in larval paddlefish ampullary organs. This suggests very close developmental and, most likely, evolutionary links between hair cells and electroreceptors. This was not necessarily to be expected: electroreceptors could have been more similar to other cell types with ribbon synapses, i.e., retinal or pineal photoreceptors, or retinal bipolar cells (indeed, Neurod4 is also important for bipolar neuron development; *Hatakeyama and Kageyama, 2004*).

The selective pressure for the evolution of electroreceptors, which enable the detection of living animals in water, is likely related to the transition from filter-feeding to predation in the lineage leading to vertebrates (*Gans and Northcutt, 1983*; *Northcutt and Gans, 1983*; *Northcutt, 2005b*). Various lines of evidence support the homology of all non-teleost electroreceptors, including those of lampreys: their stimulation by weak, low-frequency cathodal fields (and inhibition by strong cathodal fields); their innervation by pre-otic *anterior* lateral afferents projecting to the dorsal octavolateral nucleus in the hindbrain; and, in jawed vertebrates, their demonstrated embryonic origin, together with neuromasts, from lateral line placodes (*Bullock et al., 1983*; *Northcutt, 1992*; *Braun, 1996*; *New, 1997*; *Baker et al., 2013*). Given this, the striking similarities in both development and physiology that we have identified in paddlefish ampullary organs are consistent with electroreceptor evolution in the vertebrate ancestor either via the diversification of hair cells that had already evolved from ancestral mechanoreceptor cells (*Duncan and Fritzsch, 2012*; *Fritzsch and Straka, 2014*), or via the diversification of an ancestral mechanoreceptor cell - with an apical primary cilium (lost in lamprey electroreceptors), microvilli, and synaptic ribbons (*Northcutt, 1986*; *Bodznick, 1989*; *New, 1997*), and using otoferlin, Vglut3 and Ca$_v$1.3 channels for synaptic transmission - into both hair cells and electroreceptors.

## Summary and perspective

Our unbiased RNA-seq approach has shed new light on molecular mechanisms underlying ampullary organ development and physiology in a non-teleost chondrostean fish. The expression in ampullary organs, as well as neuromasts, of key hair cell transcription factor genes, such as *Atoh1*, Sox2 and *Pou4f3*, and genes encoding proteins specifically required for glutamate release at the hair cell ribbon synapse, such as otoferlin, Vglut3 and Ca$_v$1.3 channels, supports close developmental and physiological, hence evolutionary, relationships between electroreceptors and hair cells. Our identification of the first-reported ampullary organ-specific genes, including the proneural transcription factor gene *Neurod4*, a beta-parvalbumin gene, and voltage-gated potassium channel subunit genes, provides novel insight into the potential molecular basis of ampullary organ-specific development and physiology. We identify Ca$_v$1.3 (with Ca$_v\beta$2) as a candidate for the apical voltage-sensing channel (indeed, Ca$_v$1.3 was recently shown to be the voltage-sensing channel in skate electroreceptors; *Bellono et al., 2017*), and K$_v$1.5 (with unusual amino acid substitutions that may affect its properties), together with K$_v\beta$3, as a candidate for the basal membrane rectifying K$_v$ channel predicted from skate ampullary organ electrophysiology. As noted above, the paddlefish has a very limited annual spawning period, so it is technically difficult to optimize e.g. CRISPR/Cas9 for targeted mutation. However, the continuing rapid advances in CRISPR/Cas9 technology, plus the development of other experimentally tractable non-teleost model systems, should eventually make it possible to test specific gene function in this and related species. Overall, our analysis has provided wide-ranging insights into molecular aspects of ampullary organ development and physiology, and an essential framework for future comparative work to determine both the function of these genes, and the extent to which they are conserved across other non-teleost electroreceptive vertebrate groups.

## Materials and methods

### Tissue and RNA isolation and Illumina sequencing

*P. spathula* embryos were purchased over multiple spawning seasons from Osage Catfisheries Inc. (Osage Beach, MO, USA) and staged according to *Bemis and Grande (1992)*. All experiments were performed in accordance with the approved institutional guidelines and regulations of the Institutional Animal Care and Use Committee of Kennesaw State University (approved protocol #12–001). Stage 46 yolk-sac larvae were preserved in RNALater (Ambion, Thermo Fisher Scientific Inc.,

Waltham, MA, USA) overnight at 4°C. Excess solution was removed and samples were stored at −80°C until processed. Opercular (lateral line organ-enriched) and fin (no lateral line organs) tissues were manually dissected and pooled from three different sets of 6–7 specimens each, yielding three biological replicates. RNA was extracted using Trizol reagent (Ambion), according to the manufacturer's protocol. RNA concentration was assessed using a Nanodrop N1000 spectrophotometer and integrity using an Agilent 2100 Bioanalyzer (Cambridge Genomic Services). Only samples with an RNA integrity number (RIN) greater than nine were used for next-generation sequencing. Illumina RNA-sequencing library preparation and sequencing were performed by The Centre for Applied Genomics, The Hospital for Sick Children, Toronto, Canada. Libraries were prepared following the standard Illumina RNA Library Prep kit and sequenced on an Illumina HiSeq 2500, using Illumina v3 chemistry, following the multiplex paired-end protocol (2 × 100 bases).

## Assembly and analysis of transcriptome

### Read QC and trimming
Reads were subjected to various quality controls, including filtering of high-quality reads based on the score value given in fastq files (FastQC version 0.10.1; http://www.bioinformatics.babraham.ac.uk/projects/fastqc/), removal of reads containing primer/adaptor sequences and trimming of read length using Trimmomatic-0.30 (*Bolger et al., 2014*).

### De novo assembly of the transcriptome
Reads were de novo assembled using Velvet version 1.2.10 (*Zerbino and Birney, 2008*) and Oases version 0.2.08 (*Schulz et al., 2012*). Velvet was run using different k-mer lengths k27–k77 in k10 increments along with other default parameters. Oases was run using the same k-mer range. Results from these assemblies were merged, again using Velvet and Oases k-mer of k27. All assemblies were performed on a server with 64 cores and 512 Gb of RAM.

### Obtaining transcript counts
Reads were mapped back to the transcriptome using Bowtie2 version 2–2.1.0 (*Langmead and Salzberg, 2012*). As reads were uncorrected, some cleaning of the SAM files was needed to remove PCR duplicates and to remove long-stretch (>85%) poly-A sequences. Transcript counts for each sample were obtained with HTseq-count (version 0.5.4p3) (*Anders et al., 2015*). A locus-to-transcript mapping file was used to collapse related transcripts and obtain locus-level counts. Output was used as input for statistical calculations.

### Differential Expression Analysis
The BioConductor package DESeq (*Anders and Huber, 2010*) was used for differential expression analysis. A p-value of <0.1 was considered significant after adjustment (multiple testing using Benjamini-Hochberg).

### BLAST Annotation
With a dataset of 189,933 contigs (>200 bp) assembled by Velvet/ Oases, transcripts were searched against chordate and invertebrate protein datasets (obtained from SwissProt and NCBI databases) using NCBI's Basic Local Alignment Search Tool BLASTX (*McGinnis and Madden, 2004*), with an expected (E)-value cut-off of ≤1E−05 to reveal sequence conservation. For results above threshold, the UniProt protein record was obtained for the top BLAST hit against each transcript locus.

### Functional Annotation and Enrichment Analysis
Gene ontology (GO) and protein domain annotations were extracted from relevant UniProt records. Enrichment analysis was performed using hypergeometric mean and Bonferonni multiple testing correction. Enriched genes were classified using GO terms according to molecular function using the web-based resource PANTHER (*Mi et al., 2016*).

RNA-seq data have been deposited in the NCBI Gene Expression Omnibus (GEO) database under accession code GSE92470.

## Phylogenetic analysis of parvalbumins

Amino acid sequences were downloaded from GenBank and aligned using the online version of MUSCLE (*Edgar, 2004*) from the EMBL-EBI server (*Li et al., 2015*). The following Parvalbumin (Pvalb) protein sequences were used: for *Callorhinchus milii*, Pvalb, thymic CPV3-like (AFP11760), Pvalb-alpha-like (AFP11872); for *Danio rerio*, Pvalb1 (AAH71552), Pvalb2 (AAH93135), Pvalb3 (AAH46001), Pvalb4 (AAH72551), Pvalb5 (AAH92666), Pvalb6 (NP_991136), Pvalb7 (NP_991137), Pvalb8 (NP_891982), Pvalb9 (NP_891983); for *Gallus gallus*, Pvalb3/thymic CPV3 (AAA17518), Pvalb-alpha (CAX32963), Pvalb thymic/Ocm2 (NP_001007478); for *Homo sapiens*, Pvalb-alpha (NP_001302461), Ocm (AAH69468); for *Mus musculus*, Pvalb-alpha (NP_038673), Ocm (NP_149028); for *Rana catesbeiana*, Pvalb3 (AAL09922), Pvalb-alpha (BAC55948), Pvalb-beta (AC051685); for *Rattus norvegicus*, Pvalb-alpha (NP_071944), Ocm (P02631); for *Salmo salar*, Pvalb-alpha (NP_001167235), Pvalb thymic/CPV3 (ACM09534), Pvalb-beta1 (NP_001117190), Pvalb-beta2 (NP_001117189).

Phylogenetic analyses were carried out using a Bayesian framework with the parallel version of MrBayes 3.2.6 (*Huelsenbeck and Ronquist, 2001*; *Ronquist and Huelsenbeck, 2003*). One major issue with analyzing highly divergent multi-gene families is finding suitable outgroups to root the tree appropriately. Instead of using an outgroup, a relaxed molecular clock with independent gamma rates was used to infer the position of the root. A 'mixed' substitution model prior for the amino acid sequences was used to allow the program to explore and sample across substitution models. To estimate posterior probabilities of all parameters, two Metropolis-coupled Markov chain Monte Carlo (MCMCMC) runs of 100 million generations were performed, sampling every 10,000 generations and discarding the first 25% as burn-in. The resulting annotated consensus tree was subsequently edited in FigTree (http://tree.bio.ed.ac.uk/software/figtree/).

## Alignments of Shaker-related voltage-gated potassium channel amino acid sequences

Amino acid sequences for Shaker-related voltage-gated potassium channels were aligned using Seaview (*Gouy et al., 2010*) with default settings. The following protein sequences were used: *Aplysia californica* Shaker-like (NP_001191634.1), *Callorhinchus milii* $K_v1.5$ (XP_007902374.1); *Danio rerio* $K_v1.5$ (XP_005171764.1); *Drosophila melanogaster* Shaker (CAA29917.1); *Gallus gallus* $K_v1.5$ (XP_015147812.1); *Homo sapiens* $K_v1.5$ (NP_002225.2), $K_v1.1$ (NP_000208.2) and $K_v1.4$ (NP_002224.1); *Latimeria chalumnae* $K_v1.5$ (XP_006013184.1); *Lepisosteus oculatus* $K_v1.5$ (XP_006633553.2); *Nematostella vectensis* Shaker4 (AFY09706.1); and *Xenopus tropicalis* $K_v1.5$ (XP_004912856.1).

## Gene cloning, in situ hybridization and immunohistochemistry

Total RNA from stage 44–46 embryos or stage 46 opercular tissue was isolated using Trizol (Invitrogen, Carlsbad, CA), as per the manufacturer's protocol. cDNA was made using the Superscript III First Strand Synthesis kit (Invitrogen, Thermo Fisher Scientific). Gene-specific primers were used under standard PCR conditions to amplify gene fragments, which were cloned into the pDrive vector (Qiagen, Manchester, UK) and individual clones verified by sequencing (Department of Biochemistry Sequencing Facility, University of Cambridge, UK). GenBank accession numbers for cloned paddlefish cDNA fragments are as follows: *Cacna1d* KY781950, *Cacnb2* KY781951, *Ctbp2* KY781952, *Kcna5* KY781953, *Kcnab3* KY781954, *Lhx3* KY781955, *Myt1* KY781956, *Neurod4* KY781957, *Otof* KY781958, *Pou4f1* KY781959, *Pou4f3* KY781960, *Pvalbβ1/Ocm* KY781961, *Pvalbβ2* KY781962, *Rims2* KY781963, *Slc17a8* KY781964, *Sox1* KY781965 and *Sox2* KY781966. Anti-sense RNA probes were synthesized using T7 or SP6 polymerases (Promega, Southampton, UK) and digoxigenin-labeled dUTPs (Roche, Basel, Switzerland).

Whole-mount in situ hybridization and immunohistochemistry was performed as described (*Modrell et al., 2011a*). The primary antibody against Sox2 was ab92494 (rabbit, 1:200–1:400; Abcam, Cambridge, UK; reported to work in skate: http://www.abcam.com/sox2-antibody-epr3131-ab92494.html/reviews/37352). A horseradish peroxidase-conjugated goat anti-rabbit secondary (Jackson ImmunoResearch Laboratories, Inc., West Grove, PA, USA) was used at 1:600. Each RNA probe/antibody was tested a minimum of two times, using at least five embryos per experimental trial.

## Acknowledgements

This work was supported by the BBSRC (BB/F00818X/1 to CVHB), the Leverhulme Trust (RPG-383 to CVHB), the Fisheries Society of the British Isles (Research Grant to MSM) and the NSF (IOS 1557857 to HHZ; IOS 1144965 to MCD). Thanks to Rachel Lyne (Cambridge Systems Biology Centre, University of Cambridge) for submitting RNA-seq data to NCBI GEO. Thanks to Peterhouse and the Department of Physiology, Development and Neuroscience at the University of Cambridge for hosting HHZ. We also thank Tatjana Piotrowski and her lab at the Stowers Institute for Medical Research (Kansas City, MO, USA) and Steve and Pete Kahrs and the Kahrs family (Osage Catfisheries, Inc.) for hosting MSM during paddlefish spawning seasons.

## Additional information

### Funding

| Funder | Grant reference number | Author |
| --- | --- | --- |
| Biotechnology and Biological Sciences Research Council | BB/F00818X/1 | Clare VH Baker |
| Leverhulme Trust | RPG-383 | Clare VH Baker |
| Fisheries Society of the British Isles | Research Grant | Melinda S Modrell |
| National Science Foundation | IOS 1557857 | Harold H Zakon |
| National Science Foundation | IOS 1144965 | Marcus C Davis |

The funders had no role in study design, data collection and interpretation, or the decision to submit the work for publication.

### Author contributions

MSM, Conceived the project with CVHB and designed the experiments, Performed all cloning, almost all in situ hybridization and all immunostaining experiments and analysis, and generated all related figures, Wrote the manuscript, with significant contributions from CVHB and HHZ; ML, Assembled the transcriptome with ARC and performed differential expression analyses, Read and commented on the manuscript; ARC, Assembled the transcriptome with ML and performed differential expression analyses, Read and commented on the manuscript; HHZ, Performed the phylogenetic and sequence analyses for Kcna5, and generated the schematic in Figure 6, Made a significant contribution to writing the manuscript; DB, Performed the phylogenetic analysis for the parvalbumin genes, Read and commented on the manuscript; ASC, Contributed in situ hybridization data for the parvalbumin genes, Read and commented on the manuscript; MCD, Contributed to the collection and maintenance of paddlefish embryos and the sequencing of the fin transcriptome, Read and commented on the manuscript; GM, Provided overall guidance for ML and ARC, Read and commented on the manuscript; CVHB, Conceived the project with MSM and helped design the experiments, Made a significant contribution to writing the manuscript

### Author ORCIDs

Marcus C Davis, http://orcid.org/0000-0002-2462-0138
Gos Micklem, http://orcid.org/0000-0002-6883-6168
Clare VH Baker, http://orcid.org/0000-0002-4434-3107

### Ethics

Animal experimentation: All experiments were performed in accordance with the approved institutional guidelines and regulations of the Institutional Animal Care and Use Committee of Kennesaw State University (approved protocol #12-001).

## Additional files

### Supplementary files

• Supplementary file 1. Excel file listing transcripts that are lateral line-enriched at least 1.85-fold (log$_2$fold 0.89).

### Major datasets

The following dataset was generated:

| Author(s) | Year | Dataset title | Dataset URL | Database, license, and accessibility information |
|---|---|---|---|---|
| Modrell MS, Lyne M, Carr AR, Zakon HH, Campbell AS, Davis MC, Micklem G, Baker CVH | 2017 | Data from: Insights into electrosensory organ development, physiology and evolution from a lateral line-enriched transcriptome | http://www.ncbi.nlm.nih.gov/geo/query/acc.cgi?acc=GSE92470 | Publicly available at the NCBI Gene Expression Omnibus (accession no: GSE92470) |

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
