## [Decision Letter]

Thank you for submitting your article "Insights into electrosensory organ development, physiology and evolution from a lateral line-enriched transcriptome" for consideration by *eLife*. Your article has been favorably evaluated by K VijayRaghavan (Senior Editor) and two reviewers, one of whom is a member of our Board of Reviewing Editors. The following individual involved in review of your submission has agreed to reveal his identity: Hernán López-Schier (Reviewer #2).

The reviewers have discussed the reviews with one another and the Reviewing Editor has drafted this decision to help you prepare a revised submission.

The two reviewers who analyzed your work highlighted its interest, the experimental rigor of the methods used and the originality of the study. This is the first in-depth transcriptomic analysis of electrosensory ampullary organs, opening up interesting possibilities for understanding the evolution of this organ and the molecular bases of its functioning.

Based on their comments, I am pleased to inform you that we would be delighted to accept your work for publication in *eLife* provided that you can provide satisfactory responses to the requests of the two reviewers.

Reviewer 1 requests the inclusion of the correct original references for the description of the molecules studied.

Reviewer 2 has outlined several points that should be easy to address or to comment on, concerning Ribeye sequences and ways of validating the critical role of Neurod4 and Atoh1 in electroreceptor development, in particular.

*Reviewer #1:*

This study reports the first in-depth transcriptomic analysis of electrosensory ampullary organs. This work meets high technical standards and the presentation (figures and text) is convincing. Important molecular information is presented concerning the components common to hair cells (consistent with their shared evolutionary origin), and components "specific" to electroreceptors.

The major findings concern:

Transcription factors and transcriptional co-activators common to electrosensory receptors and hair cells (e.g. Atoh1, Pou4f3, members of the six and *eya* families), and one, Neurod4, not expressed by hair cells, which therefore emerges as a candidate orchestrator of the differentiation of electroreception function;

The conservation, between electroreceptors and hair cells, of the core exocytosis machinery, consistent with the morphological (ribbons) and functional characteristics common to their synapses;

Insight into the evolution of parvalbumin genes;

The development of a plausible molecular scenario underlying electrosensory transduction and associated glutamate release signaling;

The evolution of the K_v_1.5 channel, although the impact of this evolution on channel activity remains unclear.

Overall, the results provide a solid basis for understanding how electroreceptors emerged during evolution, potentially clarify the molecular bases of both electroreceptor-specific functions and functions common to hair cells and electroreceptors.

I have only a few comments:

For a broad readership, it would be interesting to clarify whether a biochemical approach to the study of electroreceptors is feasible.

For ribbon synapses, the molecular comparison should be extended to photoreceptor cells (pineal gland cells if possible).

The physiological characteristics of electroreceptors are well presented and relevant references are cited. By contrast, the original findings are not cited or are incorrectly cited for the molecular characteristics of hair cells. For instance, Yasunaga et al. (1999) relates to otoferlin rather than vglut3 and the role of otoferlin function in synaptic exocytosis was described by Roux et al. in Cell (2006). Similar errors apply to the references for Ca_v_1.3 and Notch. Careful revision of the bibliography is therefore required.

*Reviewer #2:*

The authors present an original study that is exclusively descriptive, providing data that support a high degree of conservation in the transcriptional profile that characterizes both ampullary organs and mechanosensory lateral-line organ. The dataset includes Atoh1, a conserved transcription factor that is essential for mechanosensory hair-cell formation. This suggests significant conservation between the molecular mechanisms underlying mechanoreceptor and electroreceptor development. This is likely to be the case, but not highly surprising given that Atonal also controls mechanosensory-organ development in the fly *Drosophila*.

The quality of the data is exceptional, and the paper is clearly written.

The authors also find that genes coding for ribbon-synapse proteins are also expressed in ampullary organs. In particular, the find *Ctbp2* (Ribeye) upon a candidate-gene approach. Paddlefish *Ctbp2* is expressed in both ampullary organs and neuromasts, which the authors use to suggest that Ribeye is likely to be a key component of synaptic ribbons in electroreceptors. This may be correct, but it is difficult to tell because I could not identify a reference to the sequence of the antisense probe (or fragment of the coding sequence used) to identify Ribeye, and differentiate it from *Ctbp2* (which is a protein widely expressed in many cell types). Did the probe include the A and B domains of Ribeye/ *Ctbp2*? Also, how do the authors know that they did not probe for *Ctbp1*?

The suggestion that Atoh1 and Neurod4 are likely to be critical for ampullary-organ/electroreceptor development is conceivable. However, without any suggestion as to how this could be functionally validated, the manuscript loses some impact.

I find the amino acid substitutions in K_v_1.5 the most interesting, and that it may alter channel properties in paddlefish very exciting. I will encourage the authors to validate this idea by attempting to test paddlefish K_v_1.5 in an heterologous system (mammalian cells or frog oocytes) or in vitro. This should be fairly straightforward and will increase the impact of the work.

---

## [Author Response]

*[…] Reviewer #1:*

*[…] I have only a few comments:*

*For a broad readership, it would be interesting to clarify whether a biochemical approach to the study of electroreceptors is feasible.*

We are not sure what we are being asked here, unfortunately, sorry! Does the reviewer mean that we should discuss the feasibility of a proteomics-type approach, in addition to the transcriptome profiling approach? We are not sure what the gain would be, and would appreciate clarification, if this is considered to be essential.

*For ribbon synapses, the molecular comparison should be extended to photoreceptor cells (pineal gland cells if possible).*

We have tried to include references where possible to retinal and pineal photoreceptors, and also to bipolar neurons, which we had not included initially.

*The physiological characteristics of electroreceptors are well presented and relevant references are cited. By contrast, the original findings are not cited or are incorrectly cited for the molecular characteristics of hair cells. For instance, Yasunaga et al. (1999) relates to otoferlin rather than vglut3 and the role of otoferlin function in synaptic exocytosis was described by Roux et al. in Cell (2006). Similar errors apply to the references for Ca_v_1.3 and Notch. Careful revision of the bibliography is therefore required.*

We are very grateful to the reviewer for spotting the inadvertent mis-placing of the Yasunaga et al. (1999) citation. We have included the Roux et al. (2006) citation and additional citations for Ca_v_1.3 and Neurod4 (we assume that *Neurod4* was meant, rather than Notch). We have also included citations to some additional reviews at appropriate places. We also noticed some places where citations were not in chronological order, and have amended this. We did not spot other errors, but if any still exist, please let us know!

*Reviewer #2:*

*The authors present an original study that is exclusively descriptive, providing data that support a high degree of conservation in the transcriptional profile that characterizes both ampullary organs and mechanosensory lateral-line organ. The dataset includes Atoh1, a conserved transcription factor that is essential for mechanosensory hair-cell formation. This suggests significant conservation between the molecular mechanisms underlying mechanoreceptor and electroreceptor development. This is likely to be the case, but not highly surprising given that Atonal also controls mechanosensory-organ development in the fly Drosophila.*

There was not necessarily any reason to assume that the molecular mechanisms underlying electroreceptor development would be so highly conserved with those underlying mechanoreceptor development. For example, electroreceptors could have been more closely related to retinal or pineal photoreceptors, or retinal bipolar neurons, which also have ribbon synapses. Indeed, *Drosophila atonal* is required for R8 photoreceptor development, and Neurod4 is important for the formation of retinal bipolar neurons. We have added a couple of sentences in the “Insights into electroreceptor evolution” section in the Discussion, to highlight these points.

*The authors also find that genes coding for ribbon-synapse proteins are also expressed in ampullary organs. In particular, the find Ctbp2 (Ribeye) upon a candidate-gene approach. Paddlefish Ctbp2 is expressed in both ampullary organs and neuromasts, which the authors use to suggest that Ribeye is likely to be a key component of synaptic ribbons in electroreceptors. This may be correct, but it is difficult to tell because I could not identify a reference to the sequence of the antisense probe (or fragment of the coding sequence used) to identify Ribeye, and differentiate it from Ctbp2 (which is a protein widely expressed in many cell types). Did the probe include the A and B domains of Ribeye/ Ctbp2? Also, how do the authors know that they did not probe for Ctbp1?*

Our apologies: we had forgotten to note in the text that the riboprobe used exclusively targets the A-domain, i.e., the expression pattern is Ribeye-specific. We now mention this in the text and the figure legend.

*The suggestion that Atoh1 and Neurod4 are likely to be critical for ampullary-organ/electroreceptor development is conceivable. However, without any suggestion as to how this could be functionally validated, the manuscript loses some impact.*

At the end of the Discussion section entitled “Atoh1 and Neurod4 are likely to be critical for ampullary organ/electroreceptor development”, we now include a paragraph describing why it will be difficult to optimize CRISPR/Cas9 for the paddlefish, and how we plan to move in the future to a more experimentally tractable sturgeon species (i.e., another non-teleost chondrostean ray-finned fish), in order to undertake functional experiments relating to electroreceptor development

*I find the amino acid substitutions in K_v_1.5 the most interesting, and that it may alter channel properties in paddlefish very exciting. I will encourage the authors to validate this idea by attempting to test paddlefish K_v_1.5 in an heterologous system (mammalian cells or frog oocytes) or* in vitro*. This should be fairly straightforward and will increase the impact of the work.*

We agree that this will be exciting work for the future however it is beyond the scope of the current manuscript. In order to understand the effect of the amino acid substitutions on the channel’s properties, we would want ideally to compare the properties of the native paddlefish Kv1.5 channel with those of the same channel with different combinations of “reversions” to the conserved K_v_1.5 sequence. Conversely, it would also be very interesting to substitute the paddlefish-specific amino acids into e.g. human K_v_1.5. We have added sentences to the Discussion section entitled “Amino acid substitutions may alter Kv1.5 channel properties in paddlefish”, to suggest these specific experiments for the future.